# Effects of Eccentric-Oriented Strength Training on Return to Sport Criteria in Late-Stage Anterior Cruciate Ligament (ACL)-Reconstructed Professional Team Sport Players

**DOI:** 10.3390/medicina59061111

**Published:** 2023-06-08

**Authors:** Marko D. M. Stojanović, Nikola Andrić, Mladen Mikić, Nikola Vukosav, Borko Vukosav, Dan-Nicolae Zolog-Șchiopea, Mircea Tăbăcar, Răzvan Marian Melinte

**Affiliations:** 1Training Expertise Lab, Faculty of Sport and Physical Education, University of Novi Sad, 21000 Novi Sad, Serbia; marko.ns.stojanovic@gmail.com (M.D.M.S.); nikolatrenaznaekspertiza18@gmail.com (N.A.); mmmikac@gmail.com (M.M.); 2Faculty of Sport and Physical Education, University of Novi Sad, 21000 Novi Sad, Serbia; 3Clinic for Orthopedic Surgery and Traumatology, Clinical Center of Vojvodina, 21000 Novi Sad, Serbia; vukosavnikola@gmail.com; 4Sports Medicine Department, Medical Clinic “ST Medicina”, 21000 Novi Sad, Serbia; borko.vukosav01@gmail.com; 5Orthopedic Department, Puls Hospital of Regina Maria Hospital, 540136 Targu Mures, Romania; zologdan@yahoo.de (D.-N.Z.-Ș.); rmmelinte@yahoo.de (R.M.M.); 6Orthopedic Surgery and Traumatology Department, Humanitas Hospital, 400001 Cluj Napoca, Romania; 7Fizionova Reahabilitation, 540136 Targu Mures, Romania; 8Orthopedic Surgery and Traumatology Department, Dimitrie Cantemir University, 540136 Targu Mures, Romania

**Keywords:** ACL rehabilitation, eccentric overload, team sport athletes

## Abstract

*Background and Objectives:* An effective post-injury training program is essential to regain performance and fulfill criteria for return to sport for team sport athletes following anterior cruciate ligament (ACL) reconstruction. The aim of this study was to compare the effects of 6 weeks of eccentric-oriented strength training vs. traditional strength training during the late-stage ACL-rehab phase on leg strength and vertical and horizontal jumping performance in professional team sport athletes. *Materials and Methods:* Twenty-two subjects (14 males, 8 females, age 19.9 ± 4.4 years, mass 77.4 ± 15.6 kg, height 182.4 ± 11.7 cm) (mean ± SD) with a unilateral reconstructed ACL (BTB graft) were included in the study. All participants enrolled in the same rehabilitation protocol prior to the training study. Players were randomly assigned to an experimental (ECC: n = 11, age 21.8 ± 4.6 years, mass 82.7 ± 16.6 kg, height 185.4 ± 12.2 cm), and a control group (CON: n = 11, age 19.1 ± 2.1 years, mass 76.6 ± 16.5 kg, height 182.5 ± 10.2 cm). Both groups underwent an equivolumed rehabilitation program, with the only difference being in strength training, which consisted of flywheel training vs. traditional strength training for the experimental and control groups, respectively. Testing was organized before and after the 6-week training programs and included isometric semi-squat tests (ISOSI-injured and ISOSU-uninjured legs), vertical jump tests (CMJ), single-leg vertical jump tests (SLJI-injured and SLJU-uninjured legs), single-leg hop tests (SLHI-injured and SLHU-uninjured legs), and triple hop tests (TLHI-injured and TLHU-uninjured legs). In addition, limb symmetry indexes were calculated for the isometric semi-squat (ISOSLSI) test, the single-leg vertical jump (SLJLSI), and the hop (SLHLSI) tests, as well as the triple-leg hop (THLLSI) test. *Results:* Main effects of time across training were observed for all dependent variables (posttest > pretest, *p* < 0.05). Significant group-by-time interactions were found for ISOSU (*p* < 0.05, ES = 2.51, very large), ISOSI (*p* < 0.05, ES = 1.78, large), CMJ (*p* < 0.05, ES = 2.23, very large), SLJI (*p* < 0.05, ES = 1.48, large), SLHI (*p* < 0.05, ES = 1.83, large), and TLHI (*p* < 0.05, ES = 1.83, large). *Conclusions:* This study suggests that eccentric-oriented strength training in late-stage ACL recovery, undertaken twice or three times weekly for 6 weeks, results in better outcomes than traditional strength training in leg strength, vertical jump ability, and single and triple hop tests with injured legs in professional team sport athletes. It seems that flywheel strength training can be recommended in late-stage ACL recovery for professional team sport athletes in order to regain recommended performance outcome levels faster.

## 1. Introduction

The anterior cruciate ligament (ACL) is a ligament responsible for knee stability primarily by restricting anterior translation of the tibia but also limiting knee joint medial and lateral rotation. Ligament is heavily engaged in activities including deceleration, change of direction, and/or jump landing actions—all movements that team sports like basketball, soccer, and handball are well saturated with. Consequently, ACL injuries are frequent in team sport athletes for both men and women [1], followed by surgical reconstruction and a prolonged period of rehabilitation.

The rehabilitation process is long and multifactorial, with return to sport (RTS), defined as a return to unrestricted training or competition, generally acknowledged as the primary outcome for athletes [2]. Although the time frame from surgery was traditionally used as the main criteria to establish whether an athlete is ready to RTS, a shift towards comprehensive functional testing has dominated in recent years [3]. ACL injury has a detrimental impact on fitness attributes, with deficits in strength, reactive strength, and power commonly reported [4]. Consequently, RTS testing typically relies on strength and power assessment, with a battery of tests to assess strength/power capacity but also symmetry between limbs [5]. The most common functional performance outcomes are lower body strength (isokinetic or isometric), single-leg hop, single-leg triple hop, single-leg triple crossover hop [6], and recently, vertical jump [7]. 

The effects of various training modalities on RTS-related performance outcomes have been extensively studied in the past decades [8,9]. However, there is no consensus regarding the content of the post-ACL rehabilitation modality or the effectiveness of these rehabilitation interventions [10]. Conventional rehab protocols revolve around muscle strength regain to reduce injury risk [11], with emerging evidence that strength alone is not a sole determinant of injury risk [11]. Mounting evidence indicates that neural deficits are likely an important contributing factor to increased injury risk [12]. Therefore, novel training modalities that could influence both muscle morphology and neural activity are constantly sought after [13]. In this context, eccentric training has emerged lately. It has been reported that eccentric overload exercises could optimize muscle fiber length [14], add sarcomeres in series [15], and increase pennation angle [16], consequently optimizing muscle hypertrophy and strength [17]. Several [18,19] but not all [20,21] studies reported eccentric training to be superior to traditional strength training for muscle mass, strength, and functional performance gains. In addition, eccentric strength training gains could also be specific to the training modality without functional improvements [22]. It has been reported [23] that 8 weeks of eccentric exercise with an uninjured limb promotes reduced neural activity in the frontal cortex with increased corticospinal and spinal reflex excitability, likely resulting in larger acute and chronic strength gains and muscle activity in the untrained (injured) limb [24]. Taken all together, eccentric-oriented training seems to be a promising tool to beneficially remodel both peripheral and central neural activity [25], enhance neuromuscular control, and likely reduce the incidence of injury.

Surprisingly, the effects of eccentric-oriented training protocols have been studied sparsely in the ACL-reconstructed population, with contradictory study findings. Lepley et al. [26,27] reported that 6 weeks of eccentric training produced large gains in quadriceps strength but not sagittal plane knee kinematics in early-stage non-athletic ACL patients. Friedmann-Bette et al.’s [28] study results showed that a 12-week eccentric training program (24 sessions in total) leads to superior quadriceps hypertrophy than conventional strength training in a sample of 37 recreational athletes. Significantly larger improvements in vertical jump (*p* = 0.012) and single-leg hop (*p* = 0.027) after 12 weeks of eccentric/concentric training vs. standard ACL rehabilitation were reported in early-stage (3 weeks post-op) ACL patients [29]. Finally, an 8-week eccentric bicycle training program was found to be similar to concentric training in improving knee flexion angle, strength, and patient-reported outcomes in early-stage ACL patients [30].

Even less data, considering eccentric-oriented training effectiveness, is available when mid- and last-stage ACL-rehab periods are targeted. Kasmi et al. [31] examined the effects of eccentric training, plyometric training, and eccentric/plyometric training on dynamic balance, the Lysholm knee scale, and the single-leg hop test in 40 elite female athletes following ACL reconstruction. Four months post-op (mid-stage of recovery), a short-term (6 weeks, two times per week) training study was conducted. Statistical analysis revealed that the eccentric/plyometric training combination yielded significantly greater improvements in all testing variables than the concentric, plyometric, or control groups. Furthermore, the eccentric training group was found to yield significantly bigger gains in the Lysholm knee scale in comparison to the control group (traditional rehabilitation training). Recently, one set of eccentric-oriented Bulgarian squats, performed two times per week for eight weeks during late-stage rehabilitation, was found to be a robust tool to improve quadriceps power in well-trained team sport athletes with ACL reconstruction [32]. Furthermore, study results revealed that strength gains are baseline-dependent, with the stronger the athletes, the smaller the gains over time. More studies on the topic seem prudent.

To the best of the author’s knowledge, there are no articles examining the effects of eccentric-oriented strength training on functional performance outcomes in team sport professional athletes during the late-stage rehab period. Thus, the motivation behind this study was to compare the effects of eccentric-oriented strength training vs. traditional strength training during the late-stage ACL-rehab phase on functional outcome measures in professional team sport athletes. We hypothesized that 6 weeks of eccentric-oriented training, implemented during late-stage ACL rehabilitation, would lead to greater improvements compared to traditional strength training on leg strength, vertical jump ability, and horizontal jumping performance in professional team sport athletes. Obtained results might suggest the clinical applicability of these training protocols during late-stage ACL rehab in team sports professionals in order to achieve a faster and more efficient recovery.

## 2. Materials and Methods

This study deployed a between-subject longitudinal design to examine the effects of eccentric-oriented vs. traditional strength training protocols on crucial return-to-sport performance outcomes in professional team sport players during the late-stage ACL rehabilitation phase.

### 2.1. Subjects

The sample for this study was selected out of 134 ACL patients who underwent rehabilitation in a rehab-specialized fitness facility (the Center of Excellence in Sport Science, Novi Sad, Serbia) between January 1, 2018, and December 31, 2022. In order to be selected for this study, subjects had to be professional team sports athletes, members of at least first-division teams in their respective sport. Subjects with previous ACL injuries or severe chondral defects were not included in the study, but meniscus repair or meniscectomy performed at the time of ACL reconstruction was tolerated. In addition, exclusion criteria were as follows: (1) fail more than 20% of all training sessions; (2) fail two consecutive sessions. The number of participants was estimated using G*Power 3.1. Power was set at 80% with an alpha level of 5%, and peak isometric force was considered a primary outcome, resulting in a sample size of 10 subjects per group. Finally, twenty-two subjects (soccer n = 8; basketball n = 9; handball n = 5; 14 males, 8 females, age range 16–30, age 19.9 ± 4.4 years, mass 77.4 ± 15.6 kg, height 182.4 ± 11.7 cm) (mean ± SD) with a unilateral reconstructed ACL (all performed with BTB grafts by the same surgeon) were included in the study. At the time of first testing, they were in the late-stage rehabilitation phase, between 5–6 (5.7 ± 0.4) months post-surgery. Participants were randomly allocated (lottery method) to the control group (CON: n = 11 (4 females), age 19.1 ± 2.1 years, mass 76.6 ± 16.5 kg, height 182.5 ± 10.2 cm) or experimental group (ECC: n = 11 (4 females), age 21.8 ± 4.6 years, mass 82.7 ± 16.6 kg, height 185.4 ± 12.2 cm). Throughout the study, the subjects were advised to maintain their nutritional habits and not take any nutritional supplements, especially protein or creatine supplements. Participants or their parent/legal guardian gave their written informed consent and were instructed to be free to ask questions on any occasion. In addition, they could withdraw from the study at any time without explanation. The ethics committee of the University of Novi Sad, Serbia, approved this study (protocol number: 122/2020).

### 2.2. Study Design

All participants were engaged in a standard rehabilitation program and supervised by two highly skilled practitioners (MSc and PhD in Sport Science with more than 10 years of experience and 200 ACL rehabilitations) until enrolled in the study (Figure 1).

During the early stage of rehab (around 12 weeks), care was taken to decrease swelling and inflammation and restore range of motion and muscle activity in the affected muscles. Running was allowed after the respective physician’s clearance, somewhere around 12 weeks post-op. Low-load strength training (open and closed chain), landing skills, and low-load plyometrics were progressively included at around 16 weeks post-surgery (mid-stage rehabilitation). All participants enrolled in five training sessions a week, 80–90 min per session. Finally, between 5 and 6 months post-op, physician clearance for first RTS testing (initial testing) with recommended test protocols (3) was received for all patients and organized at the earliest occasion. After initial testing, each participant was randomized to the experimental or control group and started with the corresponding rehabilitation program. Final testing, identical to the initial one, was conducted five to seven days after the intervention period. All tests were performed by an experienced strength and conditioning coach who was blinded to the present study protocol design. In addition, tests were performed at the same time of day (16:00 p.m.–18:00 p.m.) and under the same environmental conditions for all subjects (22 °C and 60% humidity). All participants were strongly instructed to abstain from any strenuous activity for at least 24 h before testing.

### 2.3. Rehabilitation Protocols

The training period was 6 weeks long, with training occurring 6 days per week for participants in both groups. As a core of the rehabilitation protocol, two to three strength training sessions were conducted per week (15 training sessions in total), with eccentric-oriented vs. traditional strength training for the ECC and CON groups, respectively (Table 1). The experimental and control groups conducted the same number of sets and repetitions per set during the study period for each strength training session. The only difference was exercise selection, with the ECC group using isoinertial devices (kBox; Exxentric AB, Bromma, Sweden) or drills that otherwise overload the eccentric phase of movement, while the CON group used traditional isotonic strength training modalities with free weights. For isoinertial exercises, two submaximal attempts were included in each set, followed by a predetermined number of maximum voluntary repetitions. A moderate load (0.075 kg m^2^) was selected according to the conclusion by Sabido et al. [33] that these loads maximize eccentric overload. In addition, subjects were instructed and strongly verbally encouraged to perform the concentric phase as fast as possible, consequently delaying the braking action to the last third of the eccentric phase in order to maximize power outputs. The CON group exercised at around 80% of one repetition maximum (1 RM).

Beyond this, participants enrolled in two/three training sessions per week consisting of medium to high aerobic load (treadmill running), upper body strength, low to moderate load agility, and deceleration/landing skills. Finally, one/two training sessions per week were dedicated to recovery procedures (Table 2).

These training sessions were identical for both groups considering drill selection, sets, reps, and rest periods, but not intensity considering the greater torque produced during eccentric contractions. All training sessions were performed in a one-on-one format by the same highly experienced strength and conditioning coach, who guided participants on how to perform each exercise. The subjects initiated each training session with a standardized warm-up on a stationary bicycle (5–7 min), followed by dynamic stretching, calisthenics, and preparatory exercises (15 min in total). The exercise load was periodized and increased progressively throughout the study period, with deloading in the third and sixth weeks.

### 2.4. Testing Procedures

#### 2.4.1. Isometric Leg Strength

The isometric leg strength test was executed on a flywheel device (D11 full, Desmotec, Biella, Italy), with peak force measured as an outcome. The participant wore a waist-fastened harness anchored to the strap and attached to the device, tightened so as not to allow vertical movement of the participant. The device has two load cells connected to a software-equipped computer (D.Soft, Desmotec, Biella, Italy). From a semi-squat position (100 degrees knee angle, hands on hips) and following a signal, the participant tries to stand upright, progressively developing maximal pressure on the plates for a total of 10 s. The measured force is read and saved on the computer. The better of the two obtained results (rest period of 2 min between trials) for both injured (ISOSI) and uninjured (ISOSU) legs, expressed in kilograms, was recorded and used in further analysis. Intra-class correlation coefficients showed excellent reliability for both ISOSI (ICC: 0.96; CI: 0.90–0.98) and ISOSU (ICC: 0.96; CI: 0.91–0.98).

#### 2.4.2. Hop Tests

For the single-leg hop test (SLHU and SLHI for uninjured and injured legs, respectively), the participant is positioned on one leg, jumps horizontally with an all-out effort, and lands on the same limb with a controlled, balanced landing. With the triple jump test (TLHU and TLHI for uninjured and injured legs, respectively), the patient is positioned on one leg, performs three consecutive horizontal jumps with an all-out effort, and lands on the same limb in a controlled manner. The hop distance for all hop tests was measured to the nearest centimeter from the starting line to the patient’s heel with a standard tape measure. Two successful trials for each limb were recorded for all tests, with the highest distances used to compute a limb symmetry index ([injured side/uninjured side;] × 100%). A limb symmetry index of <100 reveals a deficit in the injured limb. Intra-class correlation coefficients showed excellent reliability for both SLHU (ICC: 0.92; CI: 0.82–0.97) and SLHI (ICC: 0.83; CI: 0.60–0.93), with a poor limb symmetry index for the single-leg hop test (limb symmetry index, SLHLSI) (ICC: 0.50; CI: 0.19–0.79). In addition, intra-class correlation coefficients showed excellent reliability for both TLHU (ICC: 0.92; CI: 0.82–0.97) and TLHI (ICC: 0.83; CI: 0.60–0.93), with an acceptable limb symmetry index for the triple-leg hop test (limb symmetry index, TLH-LSI) (ICC: 0.78; CI: 0.48–0.90).

#### 2.4.3. Vertical Jump Tests

To perform a countermovement jump (CMJ), subjects were instructed to start with hands on the hips in an upright standing position, swiftly flex their knees to a semi-squat position, and immediately jump upward as high as possible while landing with knees extended. A contact mat (Just Jump, Probotics, Huntsville, AL, USA) measures the flight time, from which the flight height in centimeters is calculated. Intra-class correlation coefficients showed excellent reliability for CMJ (ICC: 0.98; CI: 0.96–0.99).

For single countermovement jumps (SLJU and SLJI for uninjured and injured legs, respectively), athletes started from an upright single-leg standing position on the contact mat with hands on the hips. After dynamically counter-moving to a self-selected depth, they jumped vertically with maximum effort and landed on the same leg. The best out of three trials for both vertical jump tests were recorded (Just Jump, Probotics, Huntsville, AL, USA) and used for further analysis. Limb symmetry index (SLLSI) was calculated. Intra-class correlation coefficients showed excellent reliability for both SLJU (ICC: 0.92; CI: 0.81–0.96) and SLJI (ICC: 0.94; CI: 0.87–0.97), with an acceptable limb symmetry index for the single-leg jump test (limb symmetry index, SLJ-LSI) (ICC: 0.71; CI: 0.31–0.88).

### 2.5. Statistical Analysis

Test results are presented as the mean ± standard deviation (SD). Before any statistical analysis, the normal distribution and homogeneity of the data were confirmed with the Shapiro-Wilk and Levenes tests, respectively. The test-retest reliability was assessed using an intraclass correlation coefficient (ICC) two-way mixed model and interpreted as follows: ≥0.9 = excellent; ≥0.8 = good; ≥0.7 = acceptable; ≥0.6 = questionable; ≥0.5 = poor; <0.5 = unacceptable [34].

A two-way ANOVA (2 × 2, group × time) was used to analyze the effects of eccentric training on the study outcomes. The percentage of change ([post value/pre value] − 1) was computed and reported for each variable. When the sphericity assumption was violated, the Greenhouse-Geisser correction was used for interpretation. Moreover, effect sizes (ES) were determined from ANOVA output by converting partial eta squared to Cohens d, with ES values considered to be either “trivial” (<0.20), ”small” (>0.2–0.6), “moderate” (>0.6–1.2), “large” (>1.2–2), or “very large” (>2). The level of significance was set at *p* ≤ 0.05. Data was processed using the SPSS statistical software package, version 20 (Chicago, IL, USA).

## 3. Results

All participants attended all training sessions without reporting any rehabilitation or test-related injuries. ECC and CON groups were similar for age (21.8 ± 4.6 vs. 19.1 ± 2.1 years), body mass (82.7 ± 16.6 vs. 76.6 ± 16.5 kg), and height (185.4 ± 12.2 vs. 182.5 ± 10.2 cm).

A two-way analysis of variance revealed significant main effects of time across training for all dependent variables (posttest > pretest, *p* < 0.05, Table 3).

A significant interaction effect was found for ISOSU and ISOSI, with very large and large effect sizes, respectively. Comparing the results of the pretest and posttest measurements, the ECC group had an improvement of 28.1% vs. 15.1% for the CON group for ISOSU, while for ISOSI, improvements were 27.1% and 18.1% for the ECC and CON groups, respectively. A significant interaction effect was found for CMJ, with a very large effect size. The experimental group and control group achieved progress of 12.9% and 6.7%, respectively. A significant interaction effect was observed for the single-leg jump-injured leg variable (SLJI) with a large effect size. Considering the percentage of improvements, 23.8% and 13.7% were reported for the ECC and CON groups, respectively. For the single-leg hop test-injured leg, the interaction effect showed statistically significant differences between groups, with a large effect size. When expressed as a percentage, the reported improvements were 23.9% and 8.1% for the ECC and CON groups, respectively. Finally, significant group-by-time interactions were found for the triple hop test-injured leg (TLHI), with a large effect size. The experimental group and control group achieved progress of 14.3% and 5.3%, respectively.

## 4. Discussion

The present investigation aimed to compare the effects of 6 weeks of eccentric-oriented vs. traditional strength training on return-to-sport outcomes in late-stage ACL rehabilitation in professional team sport athletes. The study results showed that, although both training programs significantly improved all tested parameters, the eccentric-oriented training program resulted in significantly greater improvements concerning lower body strength, vertical jump performance, single-leg jump with injured leg (SLJI), single-leg hop with injured leg (SLHI), and triple-leg hop with injured leg (TLHI) performance. No significant training-effect differences were determined for isometric strength (limb symmetry index), single-leg jump test (uninjured leg), single-leg jump test (limb symmetry index), single-leg hop test (uninjured leg), single-leg hop test (limb symmetry index), triple-leg hop test (uninjured leg), and triple-leg hop test (limb symmetry index). Thus, eccentric-oriented training can be considered a worthy functional performance-improvement training method in professional team sports for late-stage ACL surgery patients. It is generally recognized that strength training is a cornerstone of every sound ACL rehabilitation program [35]. It has previously been observed that eccentric-oriented training produces unique neural patterns and greater muscle mechanical tension, consequently optimizing the neuromuscular response to strength training [17]. In addition, the efficacy of eccentric training has been reported in several studies involving early-stage ACL reconstruction patients [24,30,36]. Few studies to date, however, have discussed the return to sport outcomes following eccentric-oriented training in a late-stage athletic population, clearly justifying the rationale of this study.

In a 6-week study by Kasmi et al. [31], twelve eccentric-oriented training sessions (with total volume ranging from 64 to 120 repetitions spread across 4 exercises) produced a meaningfully better improvement (*p* < 0.05–0.001) than traditional rehabilitation protocols for one-leg vertical jump and RTS hop tests in 40 elite female athletes with reconstructed ACLs. All subjects were in a mid-stage rehabilitation phase. In addition, reported data suggests that, compared to equivolumed eccentric, traditional, or plyometric training, the combination of eccentric and plyometric load was most effective in improving dynamic stability, the Lysholm knee scale, the return to sport index, and hop tests (limb symmetry indexes). The effects of eccentric training (flywheel) on maximal strength, quadriceps rate of force development (RFD), and voluntary activation were examined in 11 collegiate athletes with unilateral ACL reconstruction [32]. During eight weeks of intervention (2 sessions per week), participants performed one all-out set of Bulgarian squats on the injured leg using a flywheel in addition to regular training practice. Reported study results revealed that eccentric-oriented strength training significantly improved RFD parameters but not muscle activation. Taken together, the aforementioned studies are in line with our study findings that eccentric-oriented training is likely to induce superior strength and power-related return to sport performance outcomes in comparison to traditional rehabilitation modalities in the athletic population. This seems to be particularly valid for subjects with pronounced deficits in strength.

Several other articles addressed the effects of eccentric training on distinct return-to-sport performance outcomes in the non-professional athlete population. Lepley et al. [26] evaluated the efficacy of combined neuromuscular electrical stimulation and eccentric training to improve strength in early-stage non-athletic ACL patients. Eccentric training was conducted two times per week, with intensity set at 60% of the eccentric one-repetition maximum. Reported results suggest that eccentric exercise improved quadriceps strength significantly better than electrostimulation therapy alone and was almost identically effective as neuromuscular electrical stimulation and eccentric exercise in combination. Interestingly, the eccentric group obtained a 22% percent change after 6 weeks of training, which is similar to our study findings (28.1% and 27.1% change for the uninvolved and involved legs, respectively). The aim of Kinikli et al.’s [29] study was to determine the functional outcomes of early inclusion of eccentric vs. concentric training in ACL surgery patients. This 12-week-long study with three training sessions per week showed no significant differences between groups in terms of flexor and extensor strength, which is contrary to our study findings. In addition, vertical jump and single-leg hop performance demonstrated significantly greater improvements in the eccentric group, which corroborates our study results. Similarly, Gerber et al. [37] revealed that early inclusion of eccentric exercises increased the hopping distance of the involved limb by a significantly greater amount in the eccentric group compared to the traditional group (*p* < 0.01), which is in line with our study findings. Recently, the effects of eccentric vs. concentric cycle training were evaluated in the early post-ACL reconstruction phase [30]. While no significant differences in quadriceps strength of the affected limb were observed (by 20 to 33%), hamstring strength increased in the eccentric group only (15.2%). The authors concluded that overall eccentric progressive eccentric cycle training was not superior to equivolumed concentric training in male nonathletic patients, which is in contrast to our study findings. 

No significant training effect between eccentric and traditional training for limb symmetry indexes was found, with several possible explanations for the obtained results. As presented in Table 3, all LSI variables had relatively low reliability, impacting study results and interpretation by likely underestimating the true effect size [38]. Second, although hop tests are a recommended type of RTS testing, there are some concerns regarding the use of the uninjured limb as a control for between-limb comparisons. Bilateral neuromuscular deficits are evidenced after ACL reconstruction, which likely lead to falsely high LSI [39]. Third, limb dominance has been shown to affect LSI indexes [40] and should be considered when interpreting LSI data, which was not the case in our study. Finally, recently reported results suggest indicators of absolute performance were superior to limb symmetry when judging the return to sport after ACLR [41].

Our study results showed greater efficacy of eccentric-oriented than traditional rehabilitation programs in injured leg performance, while no differences between training programs for uninjured leg performance were shown. Mechanisms that are likely responsible for the obtained results should be concisely hypothesized. First, continuing use of eccentric exercise is able to improve muscle morphology, with increases in fascicle length and cross-sectional area while targeting type II fibers being regularly established [42]. Second, it seems that specific neural adaptations to eccentric-oriented strength training are largely responsible for the reported efficiency of this training modality. Indeed, chronic neural deficits [43,44] have been shown to prevail for years after ACL surgery [42] and likely prevent effective strengthening and retard rehabilitation considerably [45]. Recently, unique neural mechanisms during eccentric contractions were demonstrated [46], with superior excitability at the motor cortex but also neural adjustment at the spinal level contributing to enhanced muscle recruitment. In addition, emerging data suggest that eccentric training likely attenuates injury-induced neural deficits by both improving cortical excitability and targeting specific motor control pathways in the brain [47]. Collectively, this physiological distinctiveness of eccentric exercise capacity to beneficially modify peripheral and central neural activity could be the answer to why, in our study, eccentric-oriented training was found to be superior to concentric training in improving distinct return-to-sport criteria in injured legs.

A few study limitations are noteworthy. First, assessment of the hamstring strength/power parameters should be included in the return to sport test battery to allow more intergroup comparison. Second, due to the nature of the intervention, it was not possible to blind the strength and conditioning coach to group allocation. Third, the small sample size in each group limits the statistical power and precludes the generalizability of study findings. Notwithstanding these limitations, the effectiveness of eccentric-oriented training on leg strength and power-related performance outcomes in late-stage ACL-rehabilitation professional team sport players is supported by the present study findings.

## 5. Conclusions

Six weeks of eccentric-oriented training with 2–3 sessions per week (15 total sessions) produces superior enhancement in lower body strength, vertical jump, single-leg vertical jump with injured leg, single-leg hop with injured leg, and triple-leg hop with injured leg to equivolumed traditional strength training in late-stage ACL patients in professional team sport athletes. Both eccentric-oriented and traditional strength training proved effective in improving isometric strength (limb symmetry index), single-leg jump test (uninjured leg), single-leg jump test (limb symmetry index), single-leg hop test (uninjured leg), single-leg hop test (limb symmetry index), triple-leg hop test (uninjured leg), and triple-leg hop test (limb symmetry index). Therefore, eccentric-oriented strength training appears to be a potent tool to produce improvements in return-to-sport performance outcomes for professional team sport athletes in late-stage ACL rehabilitation.

## Figures and Tables

**Figure 1 medicina-59-01111-f001:**
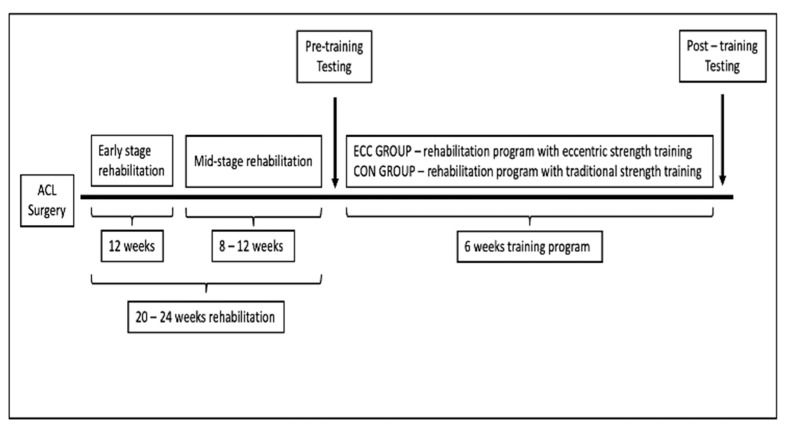
Experimental protocol for the present study.

**Table 1 medicina-59-01111-t001:** Strength training program for the EXP and CON groups.

Group	Exercises	First Week	Second Week	Third Week	Fourth Week	Fifth Week	Sixth Week
ECC eccentric oriented	Half squat on an isoinertial device	2 × 6	2 × 8	2 × 10	3 × 8	3 × 9	3 × 10
	Copenhagen eccentric	2 × 6	2 × 8	2 × 10	3 × 8	3 × 9	3 × 10
	Romanian deadlifts on isoinertial device	2 × 6	2 × 8	2 × 10	3 × 8	3 × 9	3 × 10
	Eccentric Swiss ball curl	2 × 6	2 × 8	2 × 10	3 × 8	3 × 9	3 × 10
	Hip thrust on an isoinertial device	2 × 6	2 × 8	2 × 10	3 × 8	3 × 9	3 × 10
	Bulgarian squats on an isoinertial device	2 × 6	2 × 8	2 × 10	3 × 8	3 × 9	3 × 10
CON traditional	Spanish squat	2 × 6	2 × 8	2 × 10	3 × 8	3 × 9	3 × 10
	Copenhagen	2 × 6	2 × 8	2 × 10	3 × 8	3 × 9	3 × 10
	Romanian deadlift with free weights	2 × 6	2 × 8	2 × 10	3 × 8	3 × 9	3 × 10
	Leg curl	2 × 6	2 × 8	2 × 10	3 × 8	3 × 9	3 × 10
	Hip thrust	2 × 6	2 × 8	2 × 10	3 × 8	3 × 9	3 × 10
	Bulgarian squat	2 × 6	2 × 8	2 × 10	3 × 8	3 × 9	3 × 10

**Table 2 medicina-59-01111-t002:** Rehabilitation program weekly structure.

Title 1	Monday	Tuesday	Wednesday	Thursday	Friday	Saturday	Sunday
Weeks 1, 2, and 3	Upper body strength; Agility/decelerations/landing skills; Aerobic endurance	Lower body strength	Recovery core; flexibility stretching	Upper body strength; Agility/decelerations/landing skills; Aerobic endurance	Lower body strength	Recovery core; flexibility stretching	Off
Weeks 4, 5, and 6	Upper body strength; Agility/decelerations/landing skills; Aerobic endurance	Lower body strength	Recovery core; flexibility stretching	Lower body strength	Upper body strength; Agility/decelerations/landing skills; Aerobic endurance	Lower body strength	Off

**Table 3 medicina-59-01111-t003:** Within-group differences, main and interaction effects, and effect sizes for selected variables.

Variables	Group	Pre-Test	Post-Test	Main Effect F (Sign)	Interaction F (Sign)	Effect Size
Strength (uninjured)	ECC	82.54 ± 23.16	105.72 ± 26.98 *†	192.81 (0.000)	31.62 (0.000)	2.51, very large
	CON	64.81 ± 11.46	74.63± 11.76 *			
Strength (injured)	ECC	77.09 ± 16.94	98.00 ± 20.58 *†	144.72 (0.000)	16.08 (0.001)	1.78, large
	CON	57.54 ± 14.48	68.00 ± 13.74 *			
Strength (LSI)	ECC	86.45 ± 6.02	92.45 ± 7.31 *	22.86 (.000)	0.122 (0.730)	0.156, trivial
	CON	82.81 ± 5.19	88.00 ± 4.25 *			
CMJ	ECC	45.92 ± 8.19	51.86 ± 8.11 *†	156.01 (0.000)	25.36 (0.000)	2.23, very large
	CON	37.60 ± 7.29	40.12 ± 7.47 *			
Single-leg jump (uninjured)	ECC	27.40 ± 4.88	30.91 ± 4.74 *	14.79 (0.001)	3.33 (0.08)	0.82, moderate
	CON	21.99 ± 4.77	23.18 ± 4.49 *			
Single-leg jump (injured)	ECC	22.73 ± 5.20	28.14 ± 5.01 *†	95.24 (0.000)	11.51 (0.003)	1.48, large
	CON	19.06 ± 3.47	21.68 ± 4.53 *			
Single-leg jump (LSI)	ECC	80.36 ± 5.29	90.27 ± 5.53 *	27.39 (0.000)	0.872 (0.362)	0.48, small
	CON	82.36 ± 11.35	89.97 ± 8.84 *			
Single-leg hop (uninjured)	ECC	159.54 ± 30.68	181.81 ± 21.52 *	48.33 (0.000)	2.33 (0.133)	0.69, moderate
	CON	139.54 ± 14.90	153.64 ± 13.61 *			
Single-leg hop (injured)	ECC	138.63 ± 23.56	171.81 ± 19.40 *†	70.00 (0.000)	16.79 (0.001)	1.83, large
	CON	139.54 ± 23.71	150.90 ± 17.00 *			
Single-leg hop (LSI)	ECC	87.54 ± 7.44	93.90 ± 5.14 *	8.21 (0.010)	2.09 (0.163)	0.64, moderate
	CON	91.09 ± 6.23	93.18 ± 5.60 *			
Triple-leg hop (uninjured)	ECC	550.00 ± 93.91	593.63 ± 77.72 *	47.26 (0.000)	0.965 (0.338)	0.43, small
	CON	455.90 ± 43.11	488.63 ± 35.82 *			
Triple-leg hop (injured)	ECC	494.54 ± 89.95	565.45 ± 80.82 *†	64.28 (0.000)	16.07 (0.001)	1.78, large
	CON	450.45 ± 56.32	474.09 ± 45.78 *			
Triple-leg hop (LSI)	ECC	90.00 ± 6.66	95.09 ± 4.61 *	23.01 (0.000)	1.53 (0.229)	0.55, moderate
	CON	92.81 ± 3.25	95.81 ± 3.54 *			

Pretest—initial test result ± standard deviation; Posttest—final test result ± standard deviation; * Significant difference (*p <* 0.05) between pretest and posttest. † Significantly different from the control group (CG) (*p* < 0.05).

## Data Availability

The data are available upon reasonable request.

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
