# Peer review of "Effects of Eccentric-Oriented Strength Training on Return to Sport Criteria in Late-Stage Anterior Cruciate Ligament (ACL)-Reconstructed Professional Team Sport Players"

_medicina, 2023, doi:10.3390/medicina59061111_

Round 1

Reviewer 1 Report

This study aimed to demonstrate that eccentric training is more optimal in the final part of an ACL rehabilitation program, where functional testing was used.

The study has a presentation of results and extensive research with longitudinal follow-up of the participants.

Some considerations to take into account to improve the text:

The introduction needs to clarify what practical problem the investigation is addressing. Additionally, it should explain why the study compared eccentric strength training to traditional strength training, given that the former is already known to be more effective. Lastly, it should highlight what new contribution the study makes to the science of rehabilitation. To achieve these objectives, it would be helpful to include a clear hypothesis in the text.

In the methods section, please provide the ICC and percentage of covariation for each of the tests: Isometric Leg, Hop Test, and Vertical Jump. If the ICC results are presented in the study, the objective of including them should be clearly stated. It is also important to specify when the ICC tests were conducted.

line 253-254: The number and information of the ethics committee must be located in the participants section

Control of evaluator bias must be addressed in the study. It is important to ensure that assessment bias is controlled for in these types of studies, and if it was not, it should be stated why it was not.

Likewise, with the ICC it is possible to observe the biological error of each participant, can this affect the results of the study?
They have taken this into account, otherwise it should also appear along with bias in the study design section.

results

In table 3 it is not clear to me, why the ICC values are provided, if it is not the objective of the study or is it?

I hope you take the recommendations in the best way and that they add more value to your work

I do not consider enough to recommend about the language

Author Response

This study aimed to demonstrate that eccentric training is more optimal in the final part of an ACL rehabilitation program, where functional testing was used.

The study has a presentation of results and extensive research with longitudinal follow-up of the participants.

Some considerations to take into account to improve the text:

The introduction needs to clarify what practical problem the investigation is addressing. Additionally, it should explain why the study compared eccentric strength training to traditional strength training, given that the former is already known to be more effective. Lastly, it should highlight what new contribution the study makes to the science of rehabilitation. To achieve these objectives, it would be helpful to include a clear hypothesis in the text.

[A]

We appreciate the reviewer’s comments. To respond, we changed introduction section, clarifying that studies comparing ecc and traditional training are not so consistent ( several references included). In addition, we highlighted that only few studies examined effects of eccentric training in ACL patient with also contradictory study findings. Lastly, we rephrased hypothesis and also added sentence explaining merit of this study findings . whole section now stand as “It has been reported  that eccentric overload exercises could optimize muscle fiber length [14], add sarcomeres in series [15] and increase pennation angle [16], consequently optimizing muscle hypertrophy and strength [17]. Several (18,19), but not all (cadore, franchi) studies  reported eccentric training to be superior than traditional strength training for muscle mass, strength and functional performance gains. In addition, although eccentric strength training gains could also be specific to the training modality without functional improvements ( wirth et al,2015)  it has been reported [20] that 8 weeks of eccentric exercise with uninjured limb promote positive brain adaptations, corticospinal and spinal reflex excitability and likely resulting in larger acute and chronic strength gains and muscle activity in the untrained (injured) limb [21]. Taken all together, eccentric-oriented training seems to be promising tool to beneficially remodels both peripheral and central neural activity [22], enhance neuromuscular control and likely reduce the incidence of injury.

Surprisingly, effects of eccentric-oriented training protocols, , have been studied sparsely in ACL- reconstructed population with contradictory study findings. Lepley et al. [24,25] reported that 6 weeks of eccentric training produced large gains in quadriceps strength but not sagittal plane knee kinematics in early stage non athlete ACL patients. Friedmann-Bette et al. [26]study results showed that 12week eccentric training program (24 sessions in total) leads to superior quadriceps hypertrophy than conventional strength training on a sample of 37 recreational athletes. . Significantly larger improvements in vertical jump (p = 0.012), and single leg-hop (p = 0.027), after 12 weeks of  eccentric/concentric training vs. standard ACL rehabilitation were reported in early stage (3weeks post op) ACL-patients [27]. Finally, 8-week of eccentric bicycle training programme was found to be similar to concentric training at improving knee flexion angle, strength and patient-reported outcomes in early- stage ACL-patients [28]..

Even less data considering eccentric-oriented training effectiveness are available when mid and last stage ACL-rehab periods are targeted. Kasmi et al. [29] examined the effects of eccentric training, plyometric training and eccentric/plyometric training on dynamic balance, Lysholm knee scale and the single leg hop test in 40 elite female athletes following ACL reconstruction. Four months postop (mid stage of recovery), a short term (6 weeks, two times per week) training study was conducted. Statistical analysis revealed that eccentric/plyometric training combination yielded significantly greater improvements in all testing variables than concentric, plyometric or control group. Furthermore, eccentric training group was found to yield significantly bigger gains in Lysholm knee scale in comparison to control group (traditional rehabilitation training). Recently, one set of eccentric-oriented Bulgarian squat, two times per week for eight weeks during late-stage rehabilitation, was found robust tool to improve quadriceps power in well-trained team sport athletes with ACL-reconstruction [30]. Furthermore, study results revealed that strength gains are baseline-dependent, with the stronger the athletes, the smaller the gains over time. More studies about the topic seems prudent  

To the best of authors knowledge, there are no articles examining the effects of eccentric-oriented strength training on functional performance outcomes in team sport professional athletes during late-stage rehab period. Thus, the motivation behind this study was to compare effects of  eccentric-oriented strength training vs. traditional strength training during late-stage ACL-rehab phase on the functional outcome measures  in professional team sport athletes.  We hypothesized that 6 weeks of  eccentric-oriented training, implemented during late-stage ACL-rehabilitation, would lead to greater improvements compared to traditional strength training on leg strength, vertical jump ability and horizontal jumping performance in professional team sport athletes. Obtained results might suggests clinical applicability of these training protocols during late stage ACL-rehab in team sport professionals in order to achieve faster and more efficient recovery. “

_______________________________________________________________________________

In the methods section, please provide the ICC and percentage of covariation for each of the tests: Isometric Leg, Hop Test, and Vertical Jump. If the ICC results are presented in the study, the objective of including them should be clearly stated. It is also important to specify when the ICC tests were conducted.

[A]

Corrected after suggestion. Information’s about ICC for each of the tests is now in methods section

_______________________________________________________________________________

line 253-254: The number and information of the ethics committee must be located in the participants section

[A]

Corrected after suggestion.

_______________________________________________________________________________

Control of evaluator bias must be addressed in the study. It is important to ensure that assessment bias is controlled for in these types of studies, and if it was not, it should be stated why it was not.

[A]

Corrected after suggestion, in study design section now stands “ All tests were performed by an experienced Strength& Conditioning coach who was blinded to the present study protocol design, at the same time of day (16:00 pm–18:00 pm) and environmental conditions for all subjects (22° C and ;60% humidity).“

_______________________________________________________________________________

Likewise, with the ICC it is possible to observe the biological error of each participant, can this affect the results of the study?
They have taken this into account, otherwise it should also appear along with bias in the study design section.

[A]

Corrected after suggestion. we deleted table 3 from results section.

_______________________________________________________________________________

In table 3 it is not clear to me, why the ICC values are provided, if it is not the objective of the study or is it?

[A]

Corrected after suggestion, we deleted table 3 from results section.

___________________________________________________________________________

Reviewer 2 Report

medicina-2376226_review

Title: Optimizing late-stage rehabilitation: Effects of eccentric-oriented strength training on return to sport criteria in late-stage ACL-reconstructed professional team sport players

Comments for Authors

Dear authors,

I have carefully read your paper which explored the effect of six weeks of eccentric-oriented strength training implemented during the final stage of the anterior cruciate ligament rehabilitation phase on variables such as leg strength, vertical jump ability and horizontal jumping performance in professional athletes and compared this intervention with a traditional strength training protocol.

Your results showed that this protocol with eccentric-oriented strength training in the final stage of anterior cruciate ligament recovery produced greater improvements in leg strength, vertical jump ability as well as single and triple hop test in the injured leg in professional team sport athletes compared to traditional strength training. These results offer to suggest the clinical applicability of this training protocol in professional team sports athletes with this injury in order to achieve a faster and more efficient recovery.

In my opinion, this is an interesting study, this work offers us data that can be taken into account to propose new effective training protocol in these patients.

I found minor issues in the methods and results sections that should be addressed to improve the paper, in my opinion.

Specific comments:

Abstract: Page 1, line 20: I suggest you add the full term “anterior cruciate ligament” at least the first time it appears before the acronym (ACL). In the same way, I suggest including the entire term in the title.

Introduction

-     The introduction is complete and adequate, you briefly present the hypothesis and the objective of the study.

-     Page 3, line 99: One of the two endpoints of the sentence must be removed

Materials and Methods

-     Page 3, line 142. Please, include information about how you randomly assigned participants to each group.

-     Page 3, line 142. Grammar mistake, please check it. (they.)

-     Page 3, lines 138-144. The information that appears in this paragraph are results, I suggest you move this information to the results section. Also, you could add information of the participants of each group according to the sex.

-    Page 3, line 143. What was the minimum age or age range to be able to participate in the study?

-     Figure 1 is really illustrative; I congratulate you for it.

-     Page 5, line 188. What is 1 RM? Although it is a well-known and common term, you should first add a brief explanation and then use the acronym.

-     Pages 3-4. Please, add references that support the information presented in the methodology section, references for the variables analysed, isometric leg strength, hop test, vertical jump tests.

-     Page 6, line 241 One of the two endpoints of the sentence must be removed.

-     Page 6, line 245 Remove the punctuation mark at the beginning of the sentence.

Results

-     How did you calculate the sample size for this study? please add this information.

-     Page 7, Table 3, lines 259-268. This information is related to the tests used in this study; you should move this information to the methodology section.

-     Page 7, lines 269-270.  What group does this information refer to?

-     In the methodology section you first refer to the experimental group as ECC group (page 3, line 144), later in tables 1 and 4 it appears as EXP group, please check this and unify the terminology in all sections of the text.

-     Page 8, Table 3. † This symbol refers to the results obtained in the analysis between the groups?

-     Page 8, I suggest you add and comment on the results related to the analysis between groups.

Discussion and Conclusions

-     The discussion section and conclusions are appropriate. The conclusions respond to the objectives of the study and are in line with the results obtained.

The article is generally well written, there are some grammar and punctuation mistakes that need to be corrected

Author Response

Abstract: Page 1, line 20: I suggest you add the full term “anterior cruciate ligament” at least the first time it appears before the acronym (ACL). In the same way, I suggest including the entire term in the title.

[A]

Corrected as suggested

_______________________________________________________________________________

Page 3, line 99: One of the two endpoints of the sentence must be removed

[A]

Corrected as suggested.

_______________________________________________________________________________

    Page 3, line 142. Please, include information about how you randomly assigned participants to each group.

[A]

Corrected as suggested, this sentence now stands as “Participants were randomly allocated (lottery method) to control (CON: n = 11, age 19.1±2.1 years, mass 76.6±16.5 kg, height 182.5±10.2 cm) or experimental group (ECC n = 11, age 21.8±4.6 years, mass 82.7±16.6 kg, height 185.4±12.2 cm ).“

_____________________________________________________________________________._

Page 3, line 142. Grammar mistake, please check it. (they.)

[A]

Corrected as suggested.

_____________________________________________________________________________

Page 3, lines 138-144. The information that appears in this paragraph are results, I suggest you move this information to the results section. Also, you could add information of the participants of each group according to the sex.

[A]

Point well taken. We left these lines in the Material and Methods section as it is general practice in the Journal. In addition, we add gender information for both groups , and now this sentence stand as “ Participants were randomly allocated (lottery method) to control (CON: n = 11 (4 females), age 19.1±2.1 years, mass 76.6±16.5 kg, height 182.5±10.2 cm) or experimental group (ECC n = 11 (4 females), age 21.8±4.6 years, mass 82.7±16.6 kg, height 185.4±12.2 cm ).“

Page 3, line 143. What was the minimum age or age range to be able to participate in the study?

[A]

Corrected as suggested. Sentence now stand as “Finally, twenty-two subjects, (soccer n= 8; basketball n= 9; handball n= 5; 14 males, 8 females, age range 16-30, age 19.9±4.4 years, mass 77.4±15.6 kg, height 182.4±11.7 cm) (mean±SD) with a unilateral reconstructed ACL (all done with BTB graft by the same surgeon) were included in the study.“

___________________________________________________________________________

Page 5, line 188. What is 1 RM? Although it is a well-known and common term, you should first add a brief explanation and then use the acronym.

[A]

Corrected as suggested.

___________________________________________________________________________

Pages 3-4. Please, add references that support the information presented in the methodology section, references for the variables analysed, isometric leg strength, hop test, vertical jump tests.

[A]

Corrected as suggested. Sentence now stands as “Finally, between 5- and 6-months post-op physician clearance for first RTS testing (initial testing) with recommended test protocols (3) was received for all patients and organized at earliest occasion.“

___________________________________________________________________________

Page 6, line 241 One of the two endpoints of the sentence must be removed.

[A]

Corrected as suggested.

___________________________________________________________________________

Page 6, line 245 Remove the punctuation mark at the beginning of the sentence. [A]

Corrected as suggested.

___________________________________________________________________________

-     

How did you calculate the sample size for this study? please add this information.

[A]

Corrected as suggested. Now stands as “The number of participants was estimated using G*Power 3.1. Power was set as 80%, with an alpha level of 5%, and peak isometric force considered a primary outcome,  resulting in a sample size of 10 subjects per group. Finally, twenty-two subjects...“

___________________________________________________________________________

     Page 7, Table 3, lines 259-268. This information is related to the tests used in this study; you should move this information to the methodology section.

[A]

After suggestion from other reviewer, we deleted this table with some of the data presented in Methods section.

___________________________________________________________________________

In the methodology section you first refer to the experimental group as ECC group (page 3, line 144), later in tables 1 and 4 it appears as EXP group, please check this and unify the terminology in all sections of the text.

[A]

Point well taken. Corrected as suggested

     Page 8, Table 3. † This symbol refers to the results obtained in the analysis between the groups? In addition, I suggest you add and comment on the results related to the analysis between groups.

[A]

Corrected after suggestion. Yes, this symbol refers to the between groups effects, with comments related to this in paragraph to follow…

Reviewer 3 Report

The manuscript is well written and the findings could be of potential interest for strength and conditioning coaches. However, the authors should make an effort to comply with the most recent research in order to present the current knowledge. The results section needs to be improved in order to ease reading and understanding of the findings. Furthermore, the discussion section is long and include some paragraphs not relevant for this study that should be discarded or amended to rely more on the present findings.

Abstract:

L 21: missing “the”

Introduction:

L 54: typing error, should be “women”

L 75: typing error, “in this context”

L 78-80: I suggest the authors to alleviate shortcuts about the effects of eccentric strength training. The superior effects of ECC strength training are not so consistent in the literature. I invite the authors to moderate their statements and mention contradictory studies about the superior effect of ECC strength training regarding increase in muscle mass, volume or strength (see for instance Cadore et al., 2014, DOI: 10.1111/sms.12186, or reviews from Franchi et al., 2017, DOI: 10.3389/fphys.2017.00447; Hody et al., 2019, DOI: 10.3389/fphys.2019.00536). Furthermore, ECC strength training gains could also be specific to the training modality without functional improvements (see for instance Wirth et al., 2015, DOI: 10.1519/JSC.0000000000000528).

L 82: brain adaptations can refer to a broad range of adaptations. I suggest the authors to mention here specifically changes in brain area activations mentioned in the cited study to precise the meaning of this sentence.

Methods

L 142: typing error, please delete.

L 166-167: the meaning of this sentence is somewhat difficult to perceived. Please rephrase to simplify the syntax and avoid misinterpretation.

L 167-170: split into two sentences to ease reading and comprehension.

L 195-196: how could you ensure similar volume and intensity between protocols? Providing the greater torque produced during eccentric contractions compared to concentric, the intensity should be superior as well as the training workload, i.e., volume.  Should be amended.

L 209-211: please detail whether participants were asked to develop maximal force as fast as possible (explosive), or progressively until maximal force plateaued. If participants were asked to start movement explosively, how did the experimenter account for the likely high peak force generated by the tensioning of the strap to define maximal voluntary force?

L 212: did you account for the reproducibility of the two trials (e.g.; >5% difference between trials) to validate the maximal force?

Results

The results section needs a careful editing to present table 4 and text with respect to the statistical analysis performed. Specifically, providing the two-way ANOVA, authors should report the F and p values from main effects and interactions. Please provide in the text the post-hoc results to highlight the significant differences.

Table 4: should add units of the metrics. Authors should consider enhancing the format of the table to ease the differentiation between conditions and the reading of the different values. I suggest adding columns, or changing the presentation, to include exact p values of the post-hoc tests from the different effects (main and interaction). I wonder if the description of the ES should be added in the table, that is ever crowded. Please insert the sign of the % difference

L 269-270: authors should present F values of the different variables and their corresponding p values to provide more information to readers. This can be achieved in a separate section of the text, or another column of table 4.

L 276-289: reporting % variation in the text in duplicate with table burden the reading. Please consider delete one report of the % change to ease reading.

Discussion

L 307-309: this sentence contradicts your problematic stated L 115-117 about the lack of knowledge for implementing eccentric modality in ACL rehab program. Please specify that the unknown concerns the late stage period, or alternatively, mention these studies in the introduction section to avoid hiding information.

L 325-326: I don’t understand which elements are confronted here (30.6 vs 20.6%).

L 328-331: this trainability effect is well known and I don’t really see what this element supports in the current study. Please amend or delete.

L 337-373: this section is long and presents numerous elements that should be addressed in the introduction. Numerous studies mentioned here have shown the superiority (or not) of eccentric strength training compared to concentric strength training. Therefore, these elements should be included in the introduction section since they represent the rationale of your study.

L 368: not all performances as noted at the beginning of the discussion section. Should be acknowledged.

L 389-391: architectural changes induced by eccentric strength training are not those described by the authors. Please correct with more recent evidences (see for instance Franchi et al., 2017, DOI: 10.3389/fphys.2017.00447).

L 394: “spinal reflexive excitability” not appropriate wording. Please amend.

L 385-410: although important for understanding potential mechanisms involved in strength gains, I doubt the relevancy of this section in this manuscript. First, no clues are provided in this manuscript that could indicate whether neural or architectural changes occurred in the present protocol. Second, the specificity of eccentric strength training needs careful interpretations about the training protocol implemented to determine the potential mechanisms involved in functional performance gains. Finally, I find that the link between the physiological mechanisms mentioned and their influence on performance (i.e.; maximal strength, power, etc…) is lacking, making this section a list of phenomena without their corresponding application. Please consider amending this section to makes it more relevant and shorter.

Typing errors need to be fixed but does not require an extensive proofreading.

Author Response

The manuscript is well written and the findings could be of potential interest for strength and conditioning coaches. However, the authors should make an effort to comply with the most recent research in order to present the current knowledge. The results section needs to be improved in order to ease reading and understanding of the findings. Furthermore, the discussion section is long and include some paragraphs not relevant for this study that should be discarded or amended to rely more on the present findings.

Specific comments:

Abstract:

L 21: missing “the”

[A]

Corrected after suggestion

_______________________________________________________________________________

Introduction:

L 54: typing error, should be “women”

[A]

Corrected after suggestion

_______________________________________________________________________________

Introduction:

L 75: typing error, “in this context”

[A]

Corrected after suggestion.

_______________________________________________________________________________

L 78-80: I suggest the authors to alleviate shortcuts about the effects of eccentric strength training. The superior effects of ECC strength training are not so consistent in the literature. I invite the authors to moderate their statements and mention contradictory studies about the superior effect of ECC strength training regarding increase in muscle mass, volume or strength (see for instance Cadore et al., 2014, DOI: 10.1111/sms.12186, or reviews from Franchi et al., 2017, DOI: 10.3389/fphys.2017.00447; Hody et al., 2019, DOI: 10.3389/fphys.2019.00536). Furthermore, ECC strength training gains could also be specific to the training modality without functional improvements (see for instance Wirth et al., 2015, DOI: 10.1519/JSC.0000000000000528).

[A]

Corrected after suggestion, now stands as “It has been reported  that eccentric overload exercises could optimize muscle fiber length [14], add sarcomeres in series [15] and increase pennation angle [16], consequently optimizing muscle hypertrophy and strength [17]. Several (18,19), but not all (20,21) studies  reported eccentric training to be superior than traditional strength training for muscle mass, strength and functional performance gains. In addition, although eccentric strength training gains could also be specific to the training modality without functional improvements (22).  It has been reported [23] that 8 weeks of eccentric exercise with uninjured limb promote reduced neural activity in the frontal cortex with increased corticospinal and spinal reflex excitability and likely resulting in larger acute and chronic strength gains and muscle activity in the untrained (injured) limb [24]. Taken all together, eccentric-oriented training seems to be promising tool to beneficially remodels both peripheral and central neural activity [25], enhance neuromuscular control and likely reduce the incidence of injury.”

_______________________________________________________________________________

L 82: brain adaptations can refer to a broad range of adaptations. I suggest the authors to mention here specifically changes in brain area activations mentioned in the cited study to precise the meaning of this sentence.

[A]

Corrected after suggestion, now stands as “It has been reported [23] that 8 weeks of eccentric exercise with uninjured limb promote reduced neural activity in the frontal cortex with increased corticospinal and spinal reflex excitability and likely resulting in larger acute and chronic strength gains and muscle activity in the untrained (injured) limb [24].”

_______________________________________________________________________________

L 142: typing error, please delete.

[A]

Corrected after suggestion.

_______________________________________________________________________________

Methods

L 166-167: the meaning of this sentence is somewhat difficult to perceived. Please rephrase to simplify the syntax and avoid misinterpretation.

[A]

Corrected after suggestion, now stands as “Final testing, identical to the initial one, was conducted five to seven days after intervention period.”

_______________________________________________________________________________

Methods

L 167-170: split into two sentences to ease reading and comprehension.

[A]

Corrected after suggestion, now stands as “All tests were performed by an experienced Strength& Conditioning coach who was blinded to the present study protocol design. In aditiion, tests were performed at the same time of day (16:00 pm–18:00 pm) and environmental conditions for all subjects (22° C and ;60% humidity).”

_______________________________________________________________________________

Methods

L 195-196: how could you ensure similar volume and intensity between protocols? Providing the greater torque produced during eccentric contractions compared to concentric, the intensity should be superior as well as the training workload, i.e., volume.  Should be amended.

[A]

Corrected after suggestion, now stands as “These training sessions were identical for both groups considering drill selection, sets, reps and rest periods, but not intensity considering greater torque produced during eccentric contractions

_______________________________________________________________________________

L 209-211: please detail whether participants were asked to develop maximal force as fast as possible (explosive), or progressively until maximal force plateaued. If participants were asked to start movement explosively, how did the experimenter account for the likely high peak force generated by the tensioning of the strap to define maximal voluntary force?

L 212: did you account for the reproducibility of the two trials (e.g.; >5% difference between trials) to validate the maximal force?

[A]

Corrected after suggestion, now stands as “From semi-squat position (100 degrees knee angle, hands on hips) and following a signal, the participant tries to stand upright progressively developing maximal pressure on the plates for the total of 10 s.

_______________________________________________________________________________

L 212: did you account for the reproducibility of the two trials (e.g.; >5% difference between trials) to validate the maximal force?

[A]

Point well taken. We didn’t account for the reproducibility of the two trials. Considering that all participants used same test/exercise in early and mid stage of rehab, we considered that thay are familiarized with the procedure.  

_______________________________________________________________________________

Results

The results section needs a careful editing to present table 4 and text with respect to the statistical analysis performed. Specifically, providing the two-way ANOVA, authors should report the F and p values from main effects and interactions. Please provide in the text the post-hoc results to highlight the significant differences.

Table 4: should add units of the metrics. Authors should consider enhancing the format of the table to ease the differentiation between conditions and the reading of the different values. I suggest adding columns, or changing the presentation, to include exact p values of the post-hoc tests from the different effects (main and interaction). I wonder if the description of the ES should be added in the table, that is ever crowded. Please insert the sign of the % difference

L 269-270: authors should present F values of the different variables and their corresponding p values to provide more information to readers. This can be achieved in a separate section of the text, or another column of table 4.

L 276-289: reporting % variation in the text in duplicate with table burden the reading. Please consider delete one report of the % change to ease reading.

[A]

Point well taken.   We corrected table 4 , exclude %change column and add main effect column.

_______________________________________________________________________________

Discussion

L 307-309: this sentence contradicts your problematic stated L 115-117 about the lack of knowledge for implementing eccentric modality in ACL rehab program. Please specify that the unknown concerns the late stage period, or alternatively, mention these studies in the introduction section to avoid hiding information.

[A]

Point well taken.   We rephrased sentences  and now stands as “ In addition, efficacy of eccentric training has been reported by several studies involving early stage ACL-reconstruction patients [21,28,34]. Few studies to date, however, have discussed the return to sport outcomes responses following eccentric oriented training in late stage athletic population, clearly justifying rationale of this study.”

_______________________________________________________________________________

L 325-326: I don’t understand which elements are confronted here (30.6 vs 20.6%).

[A]

Thanks for this comment, we rephrased sentence  and now stands as “ Reported study results revealed that eccentric-oriented strength training significantly improved RFD parameters  but not the muscle activation.”

_______________________________________________________________________________

L 328-331: this trainability effect is well known and I don’t really see what this element supports in the current study. Please amend or delete.

[A]

Thanks for this comment, we deleted this.

_______________________________________________________________________________

L 337-373: this section is long and presents numerous elements that should be addressed in the introduction. Numerous studies mentioned here have shown the superiority (or not) of eccentric strength training compared to concentric strength training. Therefore, these elements should be included in the introduction section since they represent the rationale of your study.

[A]

Thanks for this comment, we rephrased the whole paragraph in order to put it within the context of our study findings. Paragraph now stands as “ Several other articles addressed effects of eccentric training on distinct Return to sport performance outcomes in non-professional athlete population. Lepley et al. [24] evaluated efficacy of combined neuromuscular electrical stimulation and eccentric training to improve strength in early-stage non-athletic ACL-patients.  Eccentric training was conducted 2 times per week with intensity set at 60% of eccentric one-repetition maximum,  Reported results suggests that eccentric exercise improved quadriceps strength significantly better than electrostimulation therapy alone and almost identically  effective as  neuromuscular electrical stimulation and eccentric exercise in combination. Interestingly, eccentric group obtained 22% percent change after 6 weeks of training, which is similar to our study findings (28,1% and 27,1% change for uninvolved and involved leg, respectively). The aim of Kinikli et al. [27] study was aimed to determine the functional outcomes of an early inclusion of eccentric vs concentric training in ACL-surgery patients. This 12-week long study with 3 training sessions per week  showed no significant differences between groups in terms of flexors and extensor strength, which is contrary to our study findings. In addition, vertical jump and single leg hop performance demonstrated significantly greater improvements in eccentric group, which corroborates our study results. Similarly, Gerber et al. [35] revealed that early inclusion of eccentric exercises increased hopping distance of the involved limb by a significantly greater amount in the eccentric group compared to the traditional group (P<.01), which is in line with our study findings. Recently, effects of eccentric vs concentric cycle training were evaluated in early post ACL-reconstruction phase [28]. While no significant differences for quadriceps strength of affected limb were observed (by 20 to 33%), hamstring strength increased in the eccentric group only (15,2%). Authors concluded that overall eccentric progressive eccentric cycle training was not superior than equivolumed concentric training in male non athletic patients, which is in contrast to our study findings.” 

_______________________________________________________________________________

L 389-391: architectural changes induced by eccentric strength training are not those described by the authors. Please correct with more recent evidences (see for instance Franchi et al., 2017, DOI: 10.3389/fphys.2017.00447).

[A]

Thanks for this comment, we rephrased the sentence and changed the reference. Sentence now stands as “First, continuing use of eccentric exercise is able of favorable improve muscle morphology, with increases in fascicle length  and cross-sectional area while targeting type II fibers being regularly established [40].”

_______________________________________________________________________________

L 394: “spinal reflexive excitability” not appropriate wording. Please amend.

[A]

Point well taken. We rephrased to “Spinal excitability”

_______________________________________________________________________________

L 385-410: although important for understanding potential mechanisms involved in strength gains, I doubt the relevancy of this section in this manuscript. First, no clues are provided in this manuscript that could indicate whether neural or architectural changes occurred in the present protocol. Second, the specificity of eccentric strength training needs careful interpretations about the training protocol implemented to determine the potential mechanisms involved in functional performance gains. Finally, I find that the link between the physiological mechanisms mentioned and their influence on performance (i.e.; maximal strength, power, etc…) is lacking, making this section a list of phenomena without their corresponding application. Please consider amending this section to makes it more relevant and shorter..

[A]

Point well taken. We rephrased to “Our study results showed greater efficacy of eccentric-oriented than traditional rehabilitation program in injured leg performance while no differences between training programs for uninjured leg performance was shown.  Mechanisms that are likely responsible for the obtained results should be concisely hypothesized. First, continuing use of eccentric exercise is able of favorable improve muscle morphology, with increases in fascicle length  and cross-sectional area while targeting type II fibers being regularly established [40]. Second, it seems that specific neural adaptations to eccentric-oriented strength training are largely responsible for reported efficiency of this training modality. Indeed,  Indeed,chronic neural deficits (41,42) have been shown to prevail for years after ACL-surgery (42) and likely prevent effective strengthening and retard rehabilitation considerably [43]. Recently,  unique neural mechanisms during eccentric contractions were demonstrated [44], with superior excitability at the motor cortex but also neural adjustment at the spinal level contributing to enhanced muscle recruitment. In addition, emerging data suggest that eccentric training likely attenuate injury-induced neural deficits by both improving cortical excitability and targeting specific motor control pathways in the brain [45].  Collectively, this physiological distinctiveness of eccentric exercise capacity to beneficially modify peripheral and central neural activity  could be the answer why in our study eccentric-oriented training is found to be superior to concentric training in improving distinct return to sport criteria in injured leg. “

_________________________________________________________

Round 2

Reviewer 3 Report

I thank the authors for considering my comments. They made significant improvements in the text to consider publication.

English is sufficient to understand the article.